# Pharmacodynamic and Toxicity Studies of 6-Isopropyldithio-2′-guanosine Analogs in Acute T-Lymphoblastic Leukemia

**DOI:** 10.3390/cancers16091614

**Published:** 2024-04-23

**Authors:** Tiantian Song, Zheming Yu, Qitao Shen, Yu Xu, Haihong Hu, Junqing Liu, Kui Zeng, Jinxiu Lei, Lushan Yu

**Affiliations:** 1Institute of Drug Metabolism and Pharmaceutical Analysis, College of Pharmaceutical Sciences, Zhejiang University, Hangzhou 310058, China; 22119089@zju.edu.cn (T.S.); 22119145@zju.edu.cn (Z.Y.); huhaihong@zju.edu.cn (H.H.); 13067822113@zju.edu.cn (K.Z.); leijinxiu@zju-jhi.com (J.L.); 2Jinhua Institute of Zhejiang University, Jinhua 321099, China; shenqitao@zju-jhi.com (Q.S.); xuyu@zju-jhi.com (Y.X.); 3The First Affiliated Hospital, School of Medicine, Zhejiang University, Hangzhou 310022, China; lpxwl2001@zju.edu.cn; 4Department of Pharmacy, Second Affiliated Hospital, School of Medicine, Zhejiang University, Hangzhou 310009, China; 5Department of Pharmacy, Shaoxing People’s Hospital, Shaoxing 312068, China; 6National Key Laboratory of Advanced Drug Delivery and Release Systems, Zhejiang University, Hangzhou 310058, China

**Keywords:** acute T-lymphoblastic leukemia, thioredoxin 1, ROS, cell apoptosis, ferroptosis

## Abstract

**Simple Summary:**

Acute T-lymphoblastic leukemia is a disease with a very low survival rate after relapse, and the search for new therapeutic agents is important for the clinical benefit of patients with acute T-lymphoblastic leukemia. Our study developed YLS010, a small molecule that is more sensitive to acute T-lymphoblastic leukemia cells, based on the structure of the invented small molecule YLS004. This novel small molecule demonstrated good antitumor activity in vivo and in vitro with acceptable toxicity. These findings provide a new avenue for the development of clinical drugs for patients with acute T-lymphoblastic leukemia.

**Abstract:**

(1) Background: The research group has developed a new small molecule, 6-Isopropyldithio-2′-deoxyguanosine analogs-YLS004, which has been shown to be the most sensitive in acute T-lymphoblastic leukemia cells. Moreover, it was found that the structure of Nelarabine, a drug used to treat acute T-lymphoblastic leukemia, is highly similar to that of YLS004. Consequently, the structure of YLS004 was altered to produce a new small molecule inhibitor for this study, named YLS010. (2) Results: YLS010 has exhibited potent anti-tumor effects by inducing cell apoptosis and ferroptosis. A dose gradient was designed for in vivo experiments based on tentative estimates of the toxicity dose using acute toxicity in mice and long-term toxicity in rats. The study found that YLS010 at a dose of 8 mg/kg prolonged the survival of late-stage acute T-lymphoblastic leukemia mice in the mouse model study. (3) Conclusions: YLS010 has demonstrated specific killing effects against acute T-lymphoblastic leukemia both in vivo and in vitro. Preclinical studies of YLS010 offer a new opportunity for the treatment of patients with acute T-lymphoblastic leukemia in clinical settings.

## 1. Introduction

Acute T-lymphoblastic leukemia (T-ALL) is a disease characterized by diffuse bone marrow infiltration by immature T-lymphoblasts. It accounts for approximately 15% of the acute lymphoblastic leukemia cases in children and 25% in adults [1,2]. The clinical treatment of T-ALL currently relies on traditional chemotherapy modalities. However, T-ALL is not very responsive to current chemotherapeutic agents and is prone to developing drug resistance [3,4]. Additionally, T-ALL has a high relapse rate in the central nervous system [5], and traditional chemotherapeutic drugs are not effective in inducing remission. Therefore, the development of novel chemotherapeutic agents with high efficacy and low toxicity is crucial for improving the clinical prognosis of T-ALL patients.

The thioredoxin (Trx) system plays a crucial role in maintaining intracellular redox homeostasis. It is composed of thioredoxin (Trx), thioredoxin reductase (TrxR), and reduced coenzyme II (NADPH) [6]. The Trx system regulates the balance of thioredoxin dithiol/disulfide bonds through TrxR activity and is involved in cellular anti-oxidative stress, transcriptional regulation, and DNA synthesis [7]. Studies have shown that TrxR/Trx plays a crucial role in promoting tumor growth and inhibiting cell apoptosis, making it a significant regulator of tumor progression [8]. In mammals, there are two isoforms of Trx: cytoplasmic-type thioredoxin 1 (Trx1) and mitochondrial-type thioredoxin 2 (Trx2). Trx is highly expressed in various cancer types, such as leukemia, hepatocellular carcinoma, lung cancer, gastric cancer, and breast cancer [9]. It is essential for maintaining the tumor phenotype. Trx expression can significantly impact the chemotherapy resistance and clinical prognosis of tumors. For instance, adult leukemia cells with a high Trx1 expression are less sensitive to Doxorubicin [10]. Additionally, Trx1 expression is closely associated with poor prognosis in patients with gastric and colorectal cancers [11]. Therefore, targeting the Trx system with anti-tumor drugs could be a potential strategy for treating malignant tumors. The compound 1-Methylpropyl-2-imidazolyl disulfide (PX-12) is a specific inhibitor of Trx1. It inactivates Trx1 by reacting with the sulfhydryl groups at positions 32 and 35 of Trx1 proteins [12]. PX-12 has demonstrated a promising safety profile and good tolerability in phase I clinical trials in advanced solid tumors [13]. Although its phase II clinical trial ended in failure due to the lack of significant clinical efficacy, its potential anti-tumor efficacy is worth exploring in depth [14].

Telomerase is crucial for maintaining chromosome stability and the integrity of telomeres [15]. It is composed of telomerase reverse transcriptase (TERT), telomerase RNA component (TERC), and several auxiliary proteins [16]. TERT utilizes TERC as a template. The process involves the reverse transcription of a single strand of DNA, followed by the hydrolysis of the RNA component in the heterozygous double strand using RNA hydrolase. This leaves behind the primer fragments, which are then used in semi-conservative replication to form DNA double strands, ultimately achieving telomere lengthening [17,18]. In normal human cells, telomeres gradually shorten with each cell division. However, cells can achieve malignant transformation by activating telomerase or upregulating its expression when stimulated by oncogenic factors [19]. Studies have confirmed that telomerase activity is detectable in approximately 85–90% of primary human cancers [20]. 6-Thio-2′-deoxyuridine (6-thio-dG) is a novel telomerase activity inhibitor that can be incorporated into newly synthesized telomeres to modify them, leading to telomere dysfunction. This can induce telomere decapitation, resulting in the fast death of tumor cells [21]. Additionally, 6-thio-dG has a good safety profile in vivo and has the potential to be a broad-spectrum anti-tumor agent.

YLS004 is a recently synthesized small molecule inhibitor that is based on an in vivo understanding of drug metabolism. It enters the cell through passive diffusion and oxidizes the dithiols of the Trx-1 protein in the Trx system to form a disulfide bond. YLS004 can be metabolized to form 6-thio-dG, which inhibits telomerase and causes telomere dysfunction. 6-thio-dG becomes cytotoxic 6-TG, which is incorporated into the DNA strand to inhibit DNA synthesis. YLS004 targets three specific areas to eliminate tumor cells (Figure 1A). Our research group has previously demonstrated that YLS004 is effective against solid tumors such as human colon cancer and melanoma [22]. Previous studies have confirmed that YLS004 can induce ROS accumulation, triggering DNA damage, activating apoptosis, and exhibiting good anti-tumor effects in solid tumors such as colorectal cancer and melanoma [22]. In this study, we investigated the anti-tumor effects of YLS004 in blood tumors as a starting point and found and synthesized a new candidate compound, YLS010. The study demonstrated that YLS010 has potent anti-tumor effects on acute T-lymphoblastic leukemia cells both in vitro and in vivo, It induces the mitochondrial apoptotic pathway of the cells through the dual inhibition of Trx1 and TERT. The inhibition of the thioredoxin system leads to the accumulation of reactive oxygen species (ROS), a reduction in the glutathione (GSH) content, and the accumulation of lipid peroxides, ultimately resulting in cell ferroptosis.

## 2. Materials and Methods

### 2.1. Chemicals and Reagents

6-isopropyl dithio-2′-dexoxyguanosine (YLS004) and 6-isopropyl dithio-2′-guanosine (YLS010) were obtained from Chemipanda Bio-Tech Co., Ltd. (Hangzhou, China). 6-Thioguanine(6-TG) was purchased from Aladdin (Shanghai, China). Nelarabine was purchased from Market-Guide Pharmaceutical & Chemical Co., Ltd. (Jiangxi, China). Anti-BAX (Abcam, Cambridge, UK, Cat# ab182733, dilution ratio: 1:2000). Anti-BCL-2(Abcam, Cambridge, UK, Cat# ab32124, dilution ratio: 1:2000). Anti-Caspase 3 (Proteintech, Chicago, IL, USA, Cat# 19677-1-AP, dilution ratio: 1:1000). Anti-Caspase9 (Proteintech, Cat# 10380-1-AP, dilution ratio: 1:1000). Anti-GPX4 (Proteintech, Cat# 67763-1-lg, dilution ratio: 1:2000). Anti-ACSL4 (Proteintech, Cat# 66617-1-lg, dilution ratio: 1:5000). Anti-GAPDH (Proteintech, Cat# 60004-1-lg, dilution ratio: 1:5000).

### 2.2. Study for Exploring the Anticancer Activity of YLS004 and YLS010

#### 2.2.1. Cell Lines and Cultures

The human blood tumor cell lines used in this study were MOLT-4, Raji, Jurkat, HUT-78, and Reh., which were purchased from the Institutes for Biochemistry and Cell Biology, Chinese Academy of Sciences (Shanghai, China). Additionally, Kasumi-1, HL-60, HEL, Skm-1, Mutz-1, U266, and KM3 cells were gifts from Zhejiang Provincial Cancer Hospital (Hangzhou, China). HUT-78 cells were cultured in IMDM medium (Gibco, San Diego, CA, USA) supplemented with 20% (*v*/*v*) fetal bovine serum (Gibco), while the remaining cells were cultured in RPMI 1640 medium (Gibco) supplemented with 10% (*v*/*v*) fetal bovine serum. Both media were additionally supplemented with penicillin (100 U/mL) and streptomycin (100 U/mL). The cells were cultured in a 37 °C incubator with 5% CO_2_ for passaging, and only logarithmic growth phase cells were used for experiments.

#### 2.2.2. In Vitro Toxicity Assays

To determine the toxicity of YLS004 and YLS010 on cells, cells in the logarithmic growth phase were diluted to a density of 8 × 10^4^/mL using the corresponding medium. The cells were then inoculated into 96-well plates at a volume of 50 μL per well, At the same time, 50 μL of the medium containing different concentrations of YLS004/YLS010/Nelarabine/6-thioguanine(6-TG) was added to the wells. Six replicate wells were set up for each group. After incubating at 37 °C with 5% CO_2_ for 48 h, 10 μL of CCK-8 detection reagent was added to each well. The plate was then incubated for an additional 4 h. The absorbance at 450 nm and 650 nm was measured using an enzyme marker, and the cell viability was calculated by subtracting the 650 nm from the 450 nm absorbance in each well. The cell viability was calculated as the absorbance of each concentration/DMSO group. The drug concentration(μM) was converted to logarithmic form and plotted as the horizontal coordinate, while the cell viability was plotted as the vertical coordinate to calculate the IC50. The acquisition of each IC50 value was repeated three times.

#### 2.2.3. Apoptosis Detection Assays

To investigate the impact of YLS010 on apoptosis in T-ALL cell lines (MOLT-4, HUT-78), cells in the logarithmic growth phase were diluted to 8 × 10^4^/mL with the corresponding medium, inoculated into six-well plates, and treated with varying concentrations of YLS010. After 48 h, the cells were collected into 2 mL centrifuge tubes following centrifugation and washed twice with 1 mL PBS. After 48 h of treatment with YLS010, the cells were collected by centrifugation into two-milliliter centrifuge tubes and washed twice with one milliliter of PBS. Next, 195 μL of Annexin V-FITC binding solution was added to each sample to resuspend the cells, following the instructions of the Annexin V-FITC Apoptosis Detection Kit (Beyotine, Shanghai, China). Then, 5 μL of Annexin V-FITC and 10 μL of propidium iodide (PI) staining solution were added. The cells were incubated at room temperature in the dark for 20 min. Finally, flow cytometry was used to detect the percentage of apoptotic cells.

#### 2.2.4. Reactive Oxygen Species (ROS) Assays

To determine the level of intracellular ROS accumulation after treatment with YLS010, T-ALL cell lines in the logarithmic growth phase were diluted to 8 × 10^4^/mL with the appropriate medium. The cells were then inoculated into six-well plates and treated with varying concentrations of YLS010 to a final volume of 2 mL. After 24 h of treatment, the cells were collected by centrifugation and washed twice with 1 mL of PBS. According to the instructions of the Reactive Oxygen Detection Kit (Beyotine, Shanghai, China), 500 μL of a final concentration of 10 μmol/L DCFH-DA probe solution diluted in serum-free medium was added to each well. The cells were then incubated for 20 min at 37 °C in an incubator with inverted mixing every 5 min. After incubation, the cells were washed three times with serum-free medium to remove any DCFH-DA probe that had not entered the cells. The cell suspension was transferred to a Corning 384-well black plate. The fluorescence of each well was detected using an enzyme marker under excitation light at 488 nm and emission light at 525 nm. Quantification was performed relative to the protein concentration. The experiments were conducted independently and in parallel three times.

#### 2.2.5. Mitochondrial Membrane Potential Assay

T-ALL cell lines in the logarithmic growth phase were added to six-well plates at a density of 8 × 10^4^/mL and treated with varying concentrations of YLS010 for 24 h. After treatment, the cells were collected by centrifugation, washed once with 1 mL of PBS, and assessed for mitochondrial membrane potential using the Mitochondrial Membrane Potential Detection Kit (JC-10) (Solarbio, Beijing, China). The cells were suspended in cell culture medium and then mixed with 0.5 mL of JC-10 staining working solution. The mixture was incubated for 20 min and then washed three times with JC-10 staining buffer. Finally, quantitative analysis was performed using flow cytometry. The experiments were conducted independently and in parallel three times.

#### 2.2.6. Lipid Peroxidation Detection

Logarithmic growth phase cells were diluted to 8 × 10^4^/mL using the corresponding medium. Different concentrations of YLS010 were added to the medium, and the mixture was inoculated into a six-well plate, bringing the final volume to 2 mL. After 48 h of drug administration, the experiment was concluded. The cells were collected by centrifugation and washed with PBS. Then, 1 μL of lipid peroxidation solution was added to 1 mL of cell culture solution per well, following the instructions of the Liperfluo-Cellular Lipid Peroxidation Detection Kit (Dojindo Laboratories, Kumamoto, Japan). The Liperfluo solution was diluted with DMSO and incubated for 30 min at 37 °C. The cells were then washed twice with serum-free medium and transferred to Corning 384-well black plates. The fluorescence intensity was measured using an enzyme labeling instrument with an excitation light of 488 nm and an emission light of 525 nm for quantitative analysis. The protein concentration was used for relative quantitative analysis. The experiments were conducted independently and in parallel three times.

#### 2.2.7. Glutathione Content Assays

To analyze the effect of YLS010 on the intracellular glutathione system, the cells were treated with varying concentrations of YLS010 for 48 h. Approximately 1 × 10^6^ cells were used in 10 cm dishes. The cells were collected through centrifugation, and their content was analyzed using GSH and GSSG assay kits from Beyotine, Shanghai, China. The protein concentration was determined using the BCA protein assay kit, also from Beyotine, Shanghai, China. The quantitative analysis was performed as the GSH content divided by the protein concentration. The experiments were conducted independently and in parallel three times.

#### 2.2.8. Western Blotting

Proteins were extracted from the cells using RIPA lysis solution, and the protein concentration was determined with the BCA kit (Beyotine, Shanghai, China). The samples were then adjusted to a consistent protein concentration using 5× Protein Sampling Buffer, boiled to denature the proteins, and stored at −80 °C. Concentrated and separator gels, each with a 15% concentration, were prepared following the instructions of the One-Step PAGE Gel Kit (Vazyme Biotech, Nanjing, China). Electrophoresis was performed at a constant voltage of 70 V for 30 min and 150 V for 90 min after sampling. The samples were then electrophoresed on a PVDF membrane (Millipore, Burlington, MA, USA) at a constant current of 200 mA for 60 min. After electrophoresis, the samples were incubated at room temperature for 2 h in a shaker with 5% skimmed milk. Then, they were washed three times with 1× TBST buffer for 10 min each time. Next, the corresponding primary antibody dilution was added to the membrane and incubated at 4 °C overnight (for 14–16 h). On the following day, the primary antibody was retrieved and washed three times with TBST, and the corresponding secondary antibody dilution was prepared using 5% skimmed milk (dilution ratio of 1:5000). A total of 5000 was added and incubated for 2 h at room temperature on a shaking table. After the incubation, it was washed three times and then imaged using a chemiluminescence instrument. The Western blotting bands were analyzed by Image J and quantified after normalization using GAPDH as the endogenous control protein. The experiments were conducted independently and in parallel three times.

#### 2.2.9. Acute Toxicity Test Study in Mice

Six-week-old ICR mice (Hangzhou Academy of Medical Sciences, Hangzhou, China) were acclimatized separately by gender for one week before undergoing a toxicity study following the guidelines for acute toxicity testing. The LD_50_ of YLS010 in mice was determined to be between 300 and 425 mg/kg in the preliminary pre-test. For the formal assay, six dose groups were established (solvent control, 307 mg/kg, 350 mg/kg, 380 mg/kg, 414 mg/kg, and 425 mg/kg), each consisting of eight mice. The mice were injected via the tail vein for 14 days, and their body weights and mortality rates were recorded daily. After 14 days, blood and serum samples were collected for blood routine and biochemical analysis. Additionally, all animals underwent gross anatomy examination, and the morphology of major organs was recorded and stained with Hematoxylin and Eosin (HE).

#### 2.2.10. Long-Term Toxicity Study in Rats

Six-week-old SD rats from Hangzhou Academy of Medical Sciences in Hangzhou, China, were acclimatized and reared for one week. Then, the study groups were established, including a blank group, a solvent control group (0.9% NaCl), a low-dose group (1 mg/kg; 1/100 LD_50_), a medium-dose group (2 mg/kg; 1/50 LD_50_), and a high-dose group (10 mg/kg; 1/10 LD_50_), in accordance with the toxicity LD_50_ data of mice determined by YLS010 and the guidelines for long-term toxicity experiments on rats. The rats received daily injections via the tail vein for 14 consecutive days. Their body weights were recorded three times a week, and their activities and deaths were observed daily. After two weeks, blood and serum were collected from the rats for routine and biochemical tests. All animals underwent gross autopsy, and major organs were collected and stained with HE staining to observe tissue changes.

#### 2.2.11. Animal Model of Acute T-Lymphoblastic Leukemia

Female NOD-*Prkdc^scid^ Il2rg*^em1^(NSG) mice were purchased from Southern Model Biological Company (Shanghai, China). After a week of acclimatization, each mouse was injected with 5 × 10^6^ MOLT-4 cells in the logarithmic growth phase via the tail vein. The mice were weighed every 2 days thereafter. Anticoagulated blood was collected weekly through the tail vein to test for hCD45^+^ positivity. The mice were randomly divided into five groups of six mice each when the content of hCD45^+^ cells in their blood was ≥1%. The groups were: the Vehicle group, Nelarabine treatment group, YLS010 low concentration group, YLS010 medium concentration group, and YLS010 high concentration group. The Vehicle group (0.9% NaCl) and groups injected with different concentrations of Nelarabine [23] (100 mg/kg) and YLS010 at low (0.5 mg/kg), medium (2 mg/kg), and high (8 mg/kg) doses were continuously treated with drugs for 5 days. The study recorded changes in the body weight and survival rates of mice in each group. The humane endpoint for the mice was the onset of lower limb paralysis. After the mice were euthanized, their spleens were weighed and photographed. All animal experiments followed institutional guidelines and were approved by Zhejiang University Animal Care and Use Committee (The ethical approval number: ZJU20170387).

### 2.3. Statistical Analyses

The study’s data and statistical analyses adhered to the recommended design and analysis guidelines for pharmacological experiments. The results were expressed as the mean ± SEM and analyzed using Prism 8.0.2. The experimental data underwent t tests and one-way analysis of variance (ANOVA), with *p* < 0.05 considered statistically significant.

## 3. Results

### 3.1. YLS004 Has Better Cytotoxicity against Acute T-Lymphoblastic Leukemia Cells

We have now expanded our investigation in this section to include blood tumors, testing YLS004’s toxicity against 12 different types of blood tumor cells that cover 9 clinical diseases in vitro. The study found that T-ALL cell lines were more sensitive to toxicity compared to other cell lines (Table 1). Further comparison of the toxicity of three drugs, YLS004, the metabolite 6-TG, and the positive control drug nelarabine, against T-ALL cell lines using in vitro toxicity assays revealed that YLS004 was more toxic to the MOLT-4 and HUT-78 cell lines than metabolite 6-TG and the positive control drug Nelarabine in vivo (Figure 1B,C).

### 3.2. The Newly Synthesized Compound YLS010 Is More Cytotoxic Than YLS004

The study found that the structure of YLS004 closely resembled that of nelarabine, a clinically used drug for T-ALL, except for the deoxyguanosine and arabinose at the 2′ position. As a result, the synthetic pathway was redesigned to produce the new compound YLS010 (Figure 2A). In our in vitro toxicity studies of YLS010, we observed that it had stronger cytotoxic effects on T-ALL cells compared to YLS004 and the positive control drug Nelarabine (Figure 2C,D). Furthermore, bone marrow extracts were obtained from patients diagnosed with acute T-lymphoblastic leukemia. The leukemia cells were extracted using Ficoll, and an in vitro toxicity assay was conducted to compare the effectiveness of three drugs. The results showed that YLS010 had a much stronger anti-tumor effect on T-ALL cells compared to YLS004 and the positive control agent, nelarabine (Figure 2B). Therefore, this study delved deeper into the mechanism of action of YLS010.

### 3.3. YLS010 Induces Apoptosis in T-ALL Cells via the Mitochondrial Pathway

Apoptosis is a type of programmed death that plays an important role in the growth and proliferation of tumor cells [24]. The Trx system plays a crucial role in maintaining the cellular redox balance. YLS010 inhibits the reduction in dithiols in thioredoxin, leading to the accumulation of oxidative substances in cells. Although most of the reactions that produce the highest levels of oxidative free radicals in mammals occur in the mitochondria [25], when the concentration of mitochondrial oxidative free radicals reaches a certain level, it can cause mitochondrial dysfunction and induce apoptosis [26]. Flow cytometry analysis after 48 h of YLS010 treatment showed a significant increase in the apoptosis ratio in T-ALL cells, as evidenced by the detection of cytochrome c release into the cytoplasm and a subsequent decrease in mitochondrial membrane potential (Figure 3A–E). After 24 h of treatment with varying concentrations of YLS010, the JC-10 probe detected changes in the mitochondrial membrane potential of T-ALL cell lines (Figure 3A–F and Appendix A). Previous studies have demonstrated that the expression of the pro-apoptotic protein Bax increases following the initiation of the apoptotic program. Additionally, the content of the apoptosis inhibitory protein Bcl-2 decreases, leading to the release of cytochrome c. This, in turn, cleaves and activates caspase-9, ultimately resulting in the activation of the final apoptosis executing protein caspase-3 and the induction of apoptosis [27,28]. Through the detection of apoptosis-related protein expression, it was found that the expression of Bax, a member of the BCL-2 family, increased, while the expression of Bcl-2 decreased. Additionally, the expression of cleaved-caspase 9/pro-caspase 9 and cleaved-caspase 3/pro-caspase 3 also increased (Figure 3G–N and Appendix A). These results suggest that YLS010 induces apoptosis in T-ALL cells through the mitochondrial pathway.

### 3.4. YLS010 Causes Ferroptosis in T-ALL Cells by Inducing Oxidative Stress

Ferroptosis [29] is a type of cell death that is characterized by oxidative stress [30], a significant decrease in intracellular glutathione levels, and a high accumulation of lipid peroxides [31]. In order to determine if ferroptosis occurs in T-ALL cells treated with YLS010, we measured the levels of reactive oxygen species (ROS), GSH, and lipid peroxides. The study results indicate that treatment with YLS010 led to a significant increase in ROS and lipid peroxide levels, as well as a decrease in intracellular GSH levels (Figure 4A–F). These changes suggest that YLS010 induced ferroptosis in T-ALL cells. Previous research has demonstrated that GPX4 plays a critical role in reducing intracellular GSH levels, and its decreased expression renders cells more vulnerable to ferroptosis [32,33]. Furthermore, ACSL4, a member of the long-chain family of acyl coenzyme A synthetase, is involved in the biosynthesis and remodeling of phosphatidylethanolamine. This activation of polyunsaturated fatty acids affects their transmembrane properties. Increased expression of ACSL4 is crucial in initiating ferroptosis [34,35]. Western blot analysis showed changes in the expression of relevant proteins during iron-induced apoptosis. Specifically, the levels of GPX4 decreased and ACSL4 expression increased in T-ALL cells following treatment with varying concentrations of YLS010 (Figure 4G–L and Appendix A). This finding supports the notion that YLS010 induces oxidative stress in T-ALL cells, ultimately leading to ferroptosis.

### 3.5. Acute Toxicity Experimental Study of YLS010 on Mice

To determine the therapeutic effect of YLS010 on an acute T-lymphoblastic leukemia mouse model, as well as the therapeutic window of the drug, we conducted acute toxicity experimental studies in mice. The LD50 of YLS010 in mice was calculated by the Bliss method to be 351.92–390.07 mg/kg. After 14 days, the body weight of mice in the 414 mg/kg group significantly decreased (Figure 5A). However, there were no significant changes in the blood routine and blood biochemistry tests of mice, except for a decrease in the number of erythrocytes and the percentage of lymphocytes and an increase in the percentage of granulocytes after YLS010 treatment (Appendix A). In the administration group, an inflammatory reaction was observed in the bodies of the mice (Figure 5B–D). Additionally, HE-staining revealed clear structures in the hearts, spleens, lungs, kidneys, and brains of the mice, with no apparent abnormalities. However, the livers of the mice in the 414 mg/kg dose group showed varying degrees of inflammatory cell infiltration and localized small focal necrosis (Figure 5E).

### 3.6. Long-Term Toxicity Experimental Study of YLS010 on Rats

Based on the LD50 results of the acute toxicity test of YLS010 in mice, the long-term toxicity of YLS010 in rats was investigated. The results showed that the body weight of rats in the high-dose group was significantly decreased after 14 days of continuous administration compared with the previous three groups (Figure 6A), and the routine blood tests and biochemical indices of the high-dose group showed more severe toxic reactions (Appendix A). Blood routine tests showed a decrease in the leukocyte content, platelet count, platelet distribution width, platelet pressure area, lymphocyte count, percentage of intermediate cells, number of granulocytes, hemoglobin, and number of intermediate cells (Figure 6B). Blood biochemistry tests showed that alkaline phosphatase, total cholesterol, serum aminotransferase, and creatine kinase decreased and urea nitrogen increased in the high-dose group (Figure 6B), indicating that YLS010 caused more severe damage to the liver and kidney tissues of rats.

HE-staining of the tissues showed that YLS010 treatment resulted in more severe tissue damage in the liver, kidney, and lung of the rats, while the heart, spleen, and stomach tissues had clear boundaries and no obvious abnormality (Figure 6C). In the liver tissue, a small amount of hepatocyte steatosis was observed in both the low-dose and medium-dose groups, and tiny rounded vacuoles (black arrows) were seen in the cytoplasm. In the high-dose group, focal necrosis of hepatocytes, hemorrhage, cytosolic consolidation of necrotic cells (yellow arrowheads), and occasional periportal connective tissue hyperplasia (blue arrowheads) were seen locally. In lung tissue, extensive alveolar wall thickening, an unclear alveolar structure, and a small amount of inflammatory cell infiltration on the alveolar wall were seen in the low-dose group (red arrows), and a small amount of pulmonary macrophage aggregation in the alveolar lumen (yellow arrows) and small foci of inflammatory cell infiltration around individual blood vessels were seen in both the intermediate- and high-dose groups (red arrows). In addition, a small amount of tubular atrophy and an unclear structure were seen locally in the renal tissue of rats in the high-dose group (yellow arrows), along with occasional connective tissue proliferation around the portal area (blue arrows). There was also atrophy with an indistinct structure (yellow arrows).

### 3.7. Pharmacodynamic Study of YLS010 in a Mouse Xenograft Model of Acute T-Lymphoblastic Leukemia

The long-term toxicity experiment of YLS010 on rats revealed a significant toxic reaction at a dosage of 10 mg/kg. Therefore, in the mouse xenotransplantation model of acute T-lymphoblastic leukemia, the dosage of YLS010 was set at 0.5 mg/kg, 2 mg/kg, and 8 mg/kg. The positive control drug, Nelarabine, was administered at a dosage of 100 mg/kg, based on previous animal experiments. The mice were divided into five groups randomly when the hCD45^+^ cell content in their blood was ≥1%, as determined by flow cytometry (Figure 7A). There was no significant difference in the hCD45^+^ content among the five groups of mice (Figure 7B). Tail vein administration was performed for five consecutive days, and the body weights of mice were recorded. No significant difference in body weights was observed in each administration group (Figure 7C), indicating that there was no significant toxic response to the three doses of YLS010. The survival curves of mice were recorded, and there was no significant difference between the low-dose and medium-dose groups of the Nelarabine group and YLS010. The high-dose group had the best effect, with a slightly longer survival time than the model group, better than the remaining four groups (Figure 7D). Additionally, the spleen-weight-to-body-weight ratio of mice in the high-dose group was significantly smaller (Figure 7E,F), indicating a better treatment effect, and the spleen weight was essentially normal.

## 4. Discussions

Trx1 and telomerase are crucial in the development of tumors and can serve as therapeutic targets for drugs. Previous studies have shown that Trx1 and TERT have anti-tumor synergistic effects. Designing and synthesizing drugs with dual targets on the same compound can prove to be effective. To achieve the desired effect of synergizing and reducing toxicity, we designed and synthesized YLS004. In previous stages, YLS004 was shown to promote the apoptosis of human colorectal cancer cells and human melanoma cells by inducing oxidative stress. To explore the potential application of YLS004 in hematological tumors, we conducted in vitro toxicity screening. The results showed that YLS004 is more toxic to acute T-lymphoblastic leukemia.

Acute T-lymphoblastic leukemia (T-ALL) is a challenging hematological malignancy to treat in clinical settings. The remission rate of T-ALL is often lower than that of B-cell acute lymphoblastic leukemia (B-ALL) under the same treatment regimen, and T-ALL is prone to relapse. The remission rate of relapsed T-ALL is very low. Nelarabine, a class of FDA-approved drugs, has been used to treat T-ALL since 2005. Nelarabine is a type of metabolic drug that the FDA approved in 2005 for treating acute T-lymphoblastic leukemia. It can specifically target and kill acute T-lymphoblastic leukemia cells, resulting in higher remission rates for patients with relapsed T-ALL. During our research, we discovered that the structures of Nelarabine and YLS004 are very similar, with the only difference being an additional hydroxyl group in the position of the five-carbon sugar. To investigate the impact of this hydroxyl group on the drug’s effectiveness, we modified the structure of YLS004 and obtained the candidate compound YLS010 in this study. In a series of experiments, it was discovered that the antitumor effect of YLS010 on acute T-lymphoblastic leukemia cell lines was superior to that of YLS004 and nelarabine. This finding motivated further investigation.

To investigate the anti-tumor activity of YLS010, we analyzed apoptotic signals in cells treated with YLS010. Our results indicate that YLS010 up-regulates the expression of the BAX protein, down-regulates the expression of the BCL-2 protein, activates caspase 9 protein cleavage, and triggers the cascade activation of the caspase 3 protein, leading to apoptotic signals. YLS010 induces T-cell tumor cell death through the mitochondrial pathway. This results in apoptosis in T-ALL cell lines.

It has been demonstrated that YLS010 inhibits thioredoxin activity. The thioredoxin system is an important participant in maintaining redox homeostasis in the cell. Intracellular reactive oxygen species will be elevated after the thioredoxin system is inhibited. The role of reactive oxygen species in iron death was investigated in the T-ALL cell line using YLS010. Subsequent experiments confirmed that YLS010 increased the levels of reactive oxygen species in the T-ALL cell line. In all cell lines, YLS010 induced oxidative stress, resulting in elevated reactive oxygen species, reduced glutathione content, and the accumulation of lipid peroxides. The results suggest that YLS010 triggers iron death in T-ALL cells.

In order to design the administration dose for in vivo pharmacodynamics experiments, the toxicity of this candidate compound was investigated, and the LD50 of YLS010 in mice was determined to be in the range of 351.92–390.07 mg/kg. Further long-term toxicity tests in rats revealed that YLS010 could produce a significant toxic response when administered at a dose of 10 mg/kg and through the species-specific conversion of the dose between mice and rats. The long-term toxicity of YLS010 in mice was approximately 20 mg/kg. The results of the preliminary toxicity test provide experimental evidence for the setting of the dose in the later in vivo efficacy test. The mouse xenograft model of acute T-lymphoblastic leukemia showed that 8 mg/kg was the effective dose for this T-ALL xeno-inhibition model, and thus, we found that the therapeutic window of this candidate is small, and it may be necessary to improve the chemical structure to reduce the toxicity of this candidate at a later stage.

## 5. Conclusions

In this study, we screened 11 hematoma cells covering 9 clinical diseases. The results showed that acute T-lymphoblastic leukemia cells were the most sensitive to the toxicity of YLS004. We subsequently found that Nelarabine, a clinically positive drug for Acute T-lymphoblastic leukemia (T-ALL), was also effective. YLS010, a small molecule derived from the structural modification of YLS004, has shown a stronger antitumor effect on the T-ALL cell line compared to YLS004 and the positive control drug Nelarabine. This suggests that YLS010 has potential as a clinical drug for T-ALL. It is important to note that all evaluations presented here are objective and based solely on experimental results.

The results indicate that YLS010, a novel small molecule designed and synthesized by our group, exhibits good anti-tumor activity both in vivo and in vitro. YLS010 triggers the apoptosis of T-ALL cell lines through the mitochondrial pathway and induces the elevation of reactive oxygen species to promote the ferroptosis of T-ALL cell lines. The experimental study of YLS010 can provide new ideas for the development of clinical drugs for acute T-lymphoblastic leukemia.

## Figures and Tables

**Figure 1 cancers-16-01614-f001:**
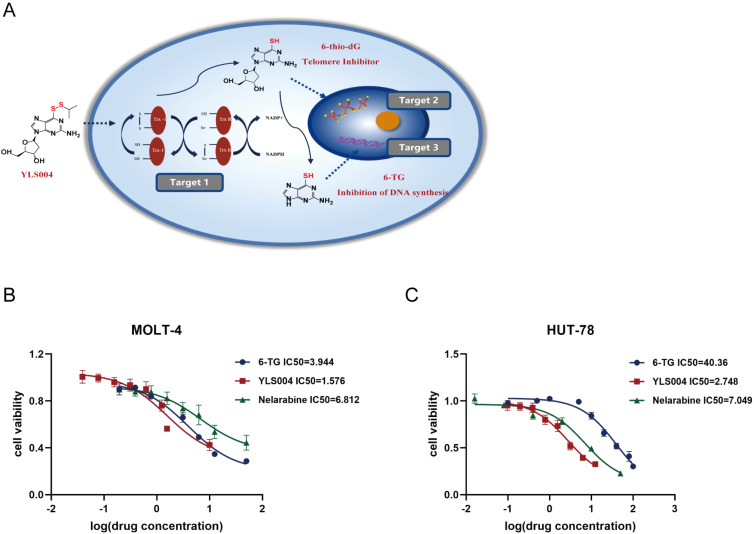
YLS004 has better cytotoxicity against acute T-lymphoblastic leukemia cells. (**A**) Diagram of the mechanism of action of YLS004. (**B**,**C**) IC50 of YLS004, metabolite 6-TG, and the positive control drug Nelarabine on MOLT-4 cells and HUT-78 cells after 48 h of treatment of these three drugs.

**Figure 2 cancers-16-01614-f002:**
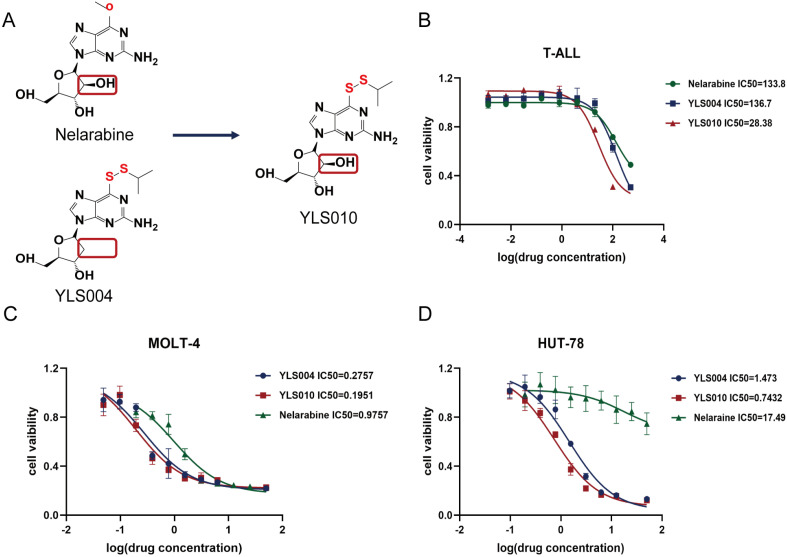
The newly synthesized compound YLS010 is more cytotoxic than YLS004. (**A**) Comparison of the chemical structures of nelarabine and YLS004 and the chemical structure of the newly synthesized compound YLS010. (**B**) BMNC cells sorted from bone marrow extracts of clinical samples of acute T-lymphoblastic leukemia were assayed for IC50 after 48 h treatment of YLS010, nelarabine, and YLS004. (**C**,**D**) In vitro toxicity assays revealed that YLS010 had a stronger toxic effect on T-ALL cell lines than YLS004 and nelarabine.

**Figure 3 cancers-16-01614-f003:**
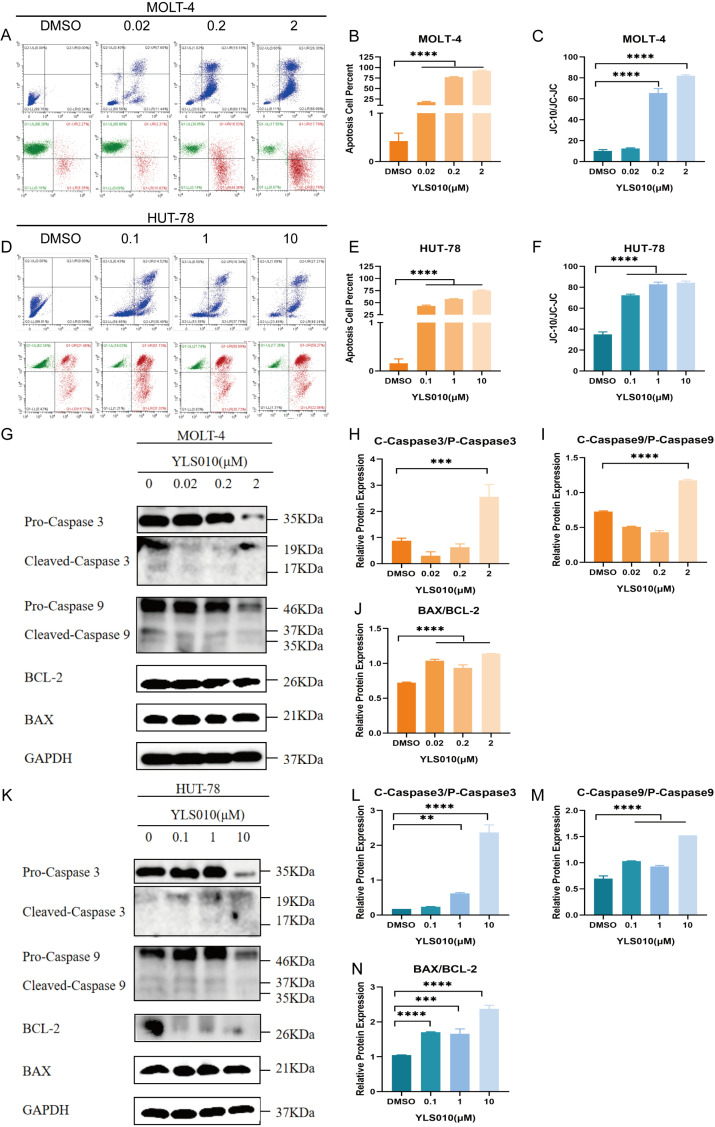
YLS010 induces apoptosis in T-ALL cells via the mitochondrial pathway. (**A**–**E**) The ratio of apoptosis and mitochondrial membrane potential changes in MOLT-4 cells and HUT-78 cells after treatment with different concentrations of YLS010 were detected using flow cytometry; (**B**,**C**,**E**,**F**) are quantitative results. (**G**–**N**) The expression levels of apoptosis-related proteins were determined using Western blotting after 48 h of treatment with different concentrations of YLS010; (**H**–**J**,**L**–**N**) are quantitative results. ** *p* < 0.01, *** *p* < 0.001, **** *p* < 0.0001.

**Figure 4 cancers-16-01614-f004:**
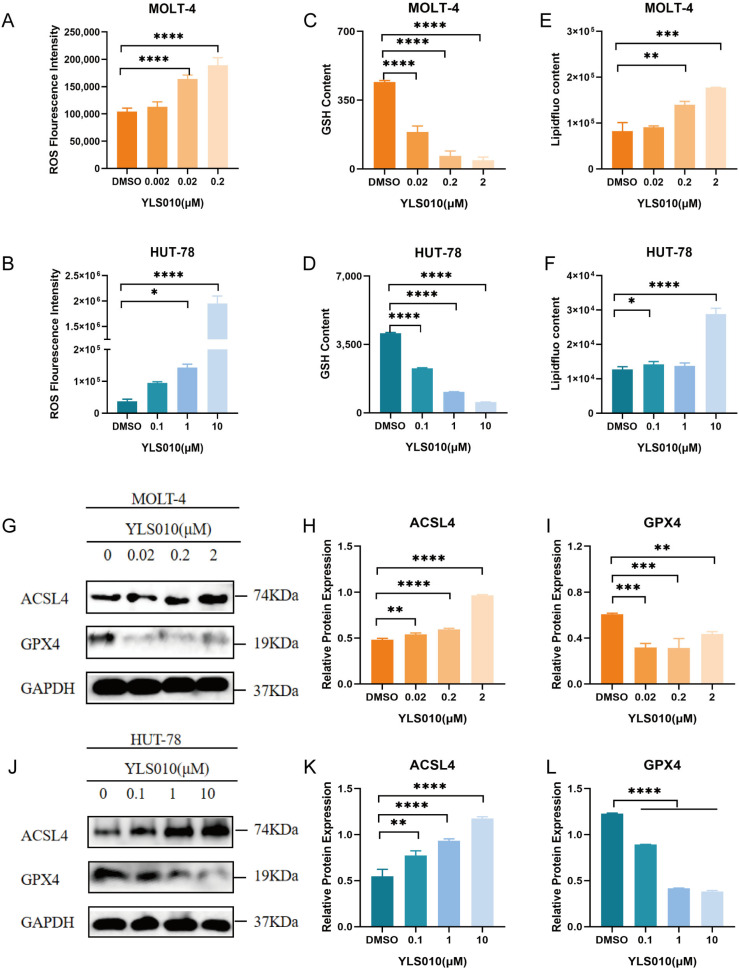
YLS010 induces ferroptosis in T-ALL cells by inducing oxidative stress. (**A**,**B**) Determination of intracellular reactive oxygen species content in T-ALL cells after 12 h of YLS010 treatment using a DCFH-DA probe. (**C**,**D**) Measurement of GSH content in T-ALL cells after 48 h of YLS010 treatment. (**E**,**F**) Detection of intracellular lipid peroxides content in T-ALL cells after the administration of YLS010 using the specific iron death probe liperfluo. (**G**–**L**) Expression levels of iron death-related proteins after treatment with different concentrations of YLS010. (**H**,**I**,**K**,**L**) Quantitative results. * *p* < 0.05, ** *p* < 0.01, *** *p* < 0.001, **** *p* < 0.0001.

**Figure 5 cancers-16-01614-f005:**
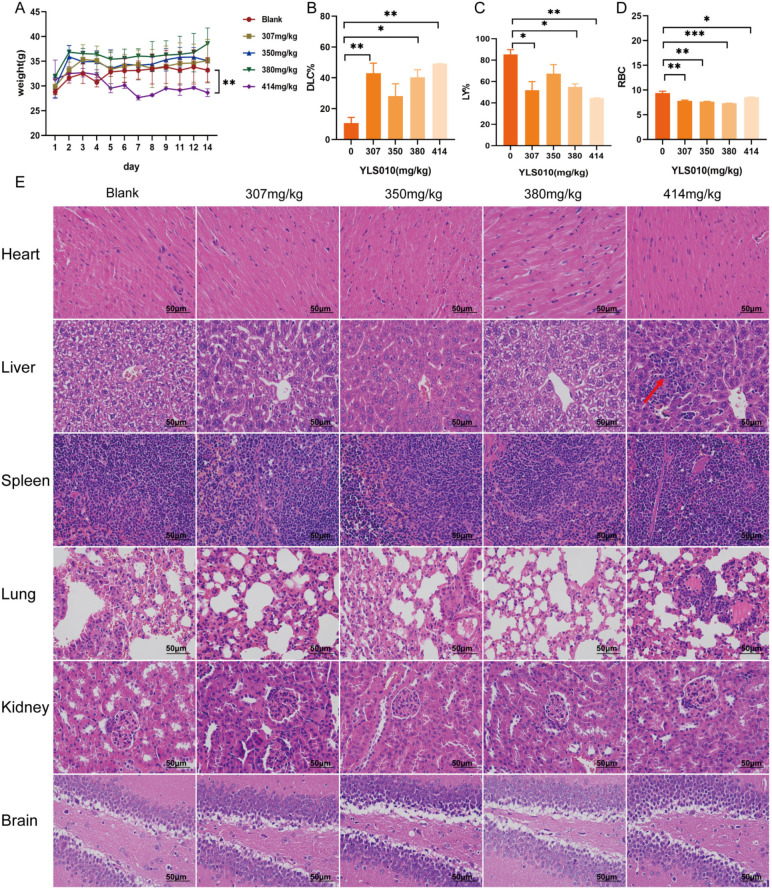
Acute toxicity experimental study of YLS010 on mice. (**A**) Change in the body weight of mice after the YLS010 tail vein injection. (**B**–**D**) Increase in the percentage of granulocytes and decrease in the percentage of erythrocytes and lymphocytes in mice after YLS010 administration. (**E**) Representative images of tissue HE-staining. Red arrows: Inflammatory cell infiltration and localized small focal necrosis. * *p* < 0.05, ** *p* < 0.01, *** *p* < 0.001.

**Figure 6 cancers-16-01614-f006:**
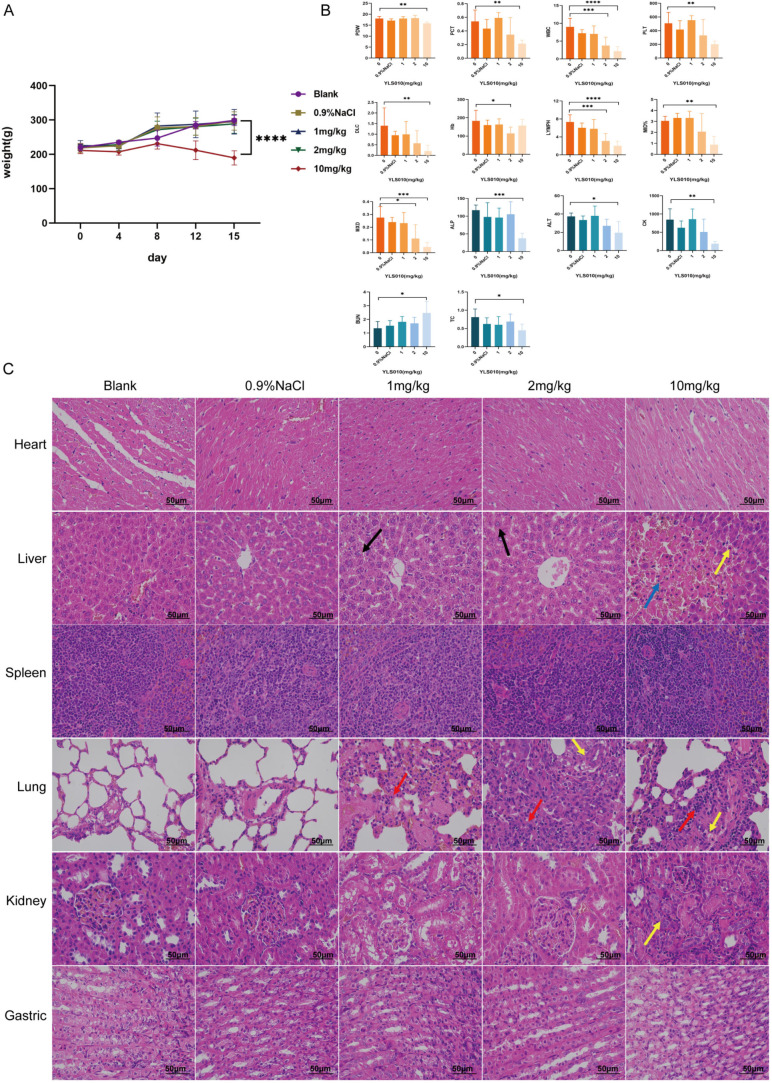
Long-term toxicity study of YLS010 in rats. (**A**) Changes in the body weight of rats in each group after 14 days of continuous administration. (**B**) Results of the blood routine test and blood biochemistry test of rats in each dose group. (**C**) HE-staining of the heart, liver, spleen, lung, kidney, and stomach of rats. Black arrows: Hepatocyte steatosis; Yellow arrows: Hemorrhage, cytosolic consolidation of necrotic cells; Red arrows: Inflammatory cell infiltration; Blue arrows: Connective tissue proliferation. * *p* < 0.05, ** *p* < 0.01, *** *p* < 0.001, **** *p* < 0.0001.

**Figure 7 cancers-16-01614-f007:**
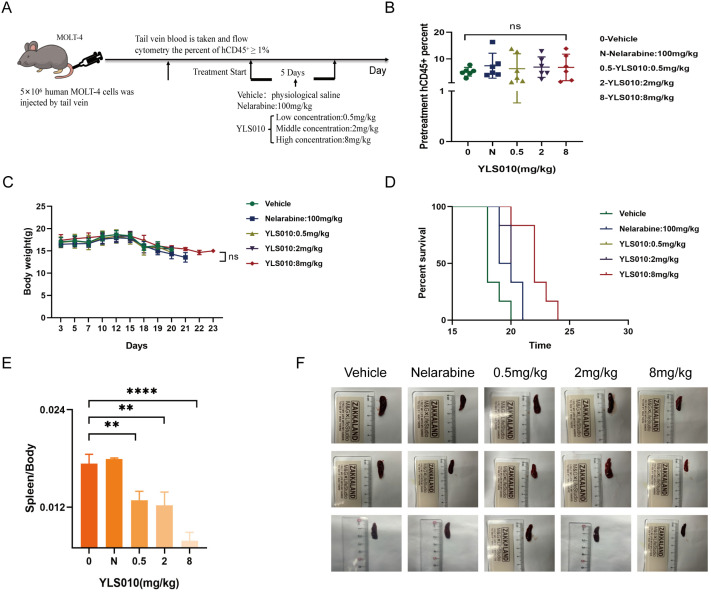
Pharmacodynamic study of YLS010 in a mouse xenograft model of acute T-lymphoblastic leukemia. (**A**) Flow of the animal experiment design. (**B**) Percentage of hCD45^+^ cells in each group of mice. (**C**) Body weight changes during the experiment in each group of mice. (**D**) Survival curves of mice in each group. (**E**) Spleen weight/body weight values of the mice in each group. (**F**) Comparison graph of the spleen size in each group of mice. ** *p* < 0.01, **** *p* < 0.0001.

**Table 1 cancers-16-01614-t001:** Toxicity sensitivity of different cell lines to YLS004.

Cell Lines	Diseases	IC_50_ (μM: SEM ± Average, n = 3)
HUT-78	Human acute T-lymphoblastic leukemia	1.84 ± 0.25
MOLT-4	Human acute T-lymphoblastic leukemia	0.56 ± 0.02
REH	Human acute non-B non-T lymphoblastic leukemia	5.34 ± 0.26
Jurkat	Human T-lymphocyte leukemia	21.8 ± 2.2
Mutz-1	Human myelodysplastic syndrome	4.11 ± 0.35
HL-60	Human promyelocytic leukemia	53.72 ± 5.61
Kasumi-1	Human acute promyelocytic leukemia	7.59 ± 0.74
Skm-1	Human acute myelogenous leukemia	3.79 ± 0.26
HEL	Human red leukocyte leukemia	30.47 ± 3.78
RAJI	Human Burkitt′s lymphoma	23.21 ± 0.92
U266	Human multiple myeloma	>100
KM3		>100

Acute T-lymphoblastic leukemia cells are the most sensitive to YLS004.

## Data Availability

The raw data supporting the conclusions of this article will be made available by the authors on request.

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
