# Peer review of "Pharmacodynamic and Toxicity Studies of 6-Isopropyldithio-2′-guanosine Analogs in Acute T-Lymphoblastic Leukemia"

_cancers, 2024, doi:10.3390/cancers16091614_

Round 1
Reviewer 1 Report
Comments and Suggestions for Authors
Song.T. and coauthors in this article analyzed the potential therapeutic power of the new small molecules, 6-Isopropyl-23 dithio-2′-deoxyguanosine analogs-YLS004 and its modified analogue YLS010, both developed by their research group. They previously demonstrated that YLS004 had good anti-tumor effects in colorectal cancer and melanoma targeting the thioredoxin and telomerase pathways by inducing ROS accumulation, triggering DNA damage and activating apoptosis.
In this paper they were interested in evaluating the effects of YLS004 molecule on different hematological malignances cell lines, finding that T-ALLs were the most sensitive to the compound. In addition YLS004 was more effective than Nelarabine, a drug used to treat acute T-lymphoblastic leukemia. Since YS004 and Nelarabina were structurally very similar, the authors decided to modify YS004 to produce the new compound YLS010 that has proven to be more effective in vitro in inducing apoptosis and ferroptosis through inhibition of Trx1 and TERT.
The authors further demonstrated the efficacy of this compound in vivo in the NSG mouse model by establishing the dose of 8 mg/kg effective in prolonging survival of mice xenofrafted with T-ALL cells with acceptable toxicity.
The topic is interesting and offers insights for future applications in therapy.
However, I have some comments:
Major comments:
1) In all experiments, both in vitro and in vivo, the number of replicates from which the mean values are obtained are never reported. Furthermore, the significant differences of the biological effects of the compounds measured as mean± SEM values should be mentioned in the results.
2) Ref 21 and 24 cited in Introduction section lane 92 and in Results section lane 282 do not relate to the topic “Our research group has previously demonstrated that YLS004 is effective against solid tumors such as human colon cancer and melanoma”.
3) The first part of Results paragraph 3.1 is redundant with the introduction. It can be summarized.
4) The experiments in paragraph 3.2 are performed with cells from bone marrow samples from patients with acute T-lymphoblastic leukemia. What is the percentage of the leukemic cells?
5) In figure 2 caption lane 311 “PBMC cells sorted from bone marrow extracts”. PBMC is for Peripheral blood mononuclear cell, it is not an abbreviation pertinent to bone marrow mononuclear cells.
6) Results paragraph 3.3 lanes 329-331 What do these statements have to do with anything? “It is important to note that the language used is clear, concise, and objective, with a formal register and precise word choice. The text follows a logical structure with causal connections between statements and adheres to conventional academic formatting and citation style.”
7) The western Blots in Figure 3C are representative of how many experiments summarized in Fig 3D?
8) The western Blot in figure 4G related to Caspase 3 doesn't look like a single membrane. By enlarging the panel, you can see a white stripe that divides the procaspase blot from the cleaved one. The same is observed in panel K. Furthermore, the bands of cleaved caspase do not appear sharp.
9) Results paragraph 3.4. Here too the first part must be summarized. The topic is more of an Introduction section than a results section.
10) Simple summary: lanes 19-20: Referring to the small molecule YLS010, the authors state that they identified a novel small molecule that specifically kills acute lymphoblastic leukemia cells. Actually, experimentally they tested the YLS004 small molecule, from which YLS010 was derived, on other in vitro tumor models and they found a greater sensitivity in acute T-lymphoblastic leukemia cells.
11) I didn’t find any mention of the Supplementary data in the manuscript.
12) Statistical methods lane 270 “The results were expressed as SEM ± mean” it should be mean± SEM
Comments on the Quality of English Language
The quality of English is fine. I recommend checking some editing carefully.
Author Response
1.Thanks for the reminder, the number of replicates for all experiments has now been added to the article. 2.Thanks for the reminder,After checking, we found that the references were cited in the wrong place, we have now checked all the references and put them in the right place. 3.In the introduction , we transfer the introduction to YLS004 from the previous paragraph of 3.1. 4.Thank you for the teacher's question, Following communication with the hospital faculty, it was found that the patient's acute T-lymphoblastic leukemia cell ratio was 89% in peripheral blood, 85% in bone marrow conventional primitive cells, and 93% in bone marrow flow. 5.Thank you for reminding us that the abbreviation for Bone Marrow Mononuclear Cell should be BMNC, which we have now revised. 6.Thank you for the reminder. We have reviewed the article and found that we were not careful enough. We have removed the reminder to the author at the end of the periodical class. Thank you again! 7.The 3D quantitative plot summarizes the results of the three experiments in 3C. 8.In Figure 4, the caspase 3 appears as two bands. As pro-caspase 3 is 35kD and cleaved-caspase 3 is 17kDa and 19kDa, the middle bands were separated and incubated individually, but from the same membrane. 9.Thank you for your suggestion. We have revised the text to summarize Part 1 in 3.5, as there is no Part 1 in 3.4. 10.Thank you for your suggestion. It has been modified. 11.The citation for the supplementary material has been updated. Please review it. 12.Thanks for the reminder,Modification completed.

Reviewer 2 Report
Comments and Suggestions for Authors
T- Acute lymphoblastic leukemia is an aggressive disease with poor prognosis e and high risk of relapse and drug resistance development after conventional chemotherapy.
Basing on the recent better understanding of leukemia cell biology and on the evidence that redox homeostasis and telomerase activity have a crucial role in blast cells survival, the authors investigated the in vitro activity of a new small molecule inhibitor, YLS010, belonging to YLS004 class compound, known to induce ROS accumulation and DNA damage and apoptosis activation in many solid tumors. They performed a series of accurate experiments to evaluate toxicity ROS production, Trx1 activity and acute and long term toxicity in rat models, concluding that the tested molecule has high activity on T-leukemia cell lines, both in vitro and in vivo, and suggest that this evidences provide the basis for future clinical studies and for the development of new molecule against T lymphoblastic leukemia.
The paper is interesting, experiment well projected, done and clearly described.
I have just few question:
1. They concluded that T-lymphoblastic leukemia is more sensitive compared to all the “other origin” cell lines to YLS004, and inferred the same efficacy for YLS010. On what basis? Among the tested cell lines there weren’t B-lymphoblastic leukemia cells. It would be interesting to know it the drug Is active in this setting.
2. Are the doses used for toxicity tests compatible with the clinical use?
3. From toxicity experiments resulted high liver, kidney and lung toxicity. Was the damage reversible?
4. Have the authors any idea on the activity/efficacy of YLS010 in “wild” leukemic samples?
Author Response
Thank you for your questions. Regarding the first question, our findings indicate that YLS004 is more toxic to acute T-lymphoblastic leukemia cells than to other blood cancers due to the ability of nelarabine to specifically kill T-cells. YLS010 is believed to be a modification of the structure of YLS004, which was derived from nelarabine. Since both molecules from which YLS010 was derived showed specific killing of acute T-lymphocytes, it is hypothesized that YLS010 may also be potentially specific for acute T-lymphocytes.
Additionally, the toxic effects of YLS004 were evaluated on the Burkitt's lymphoma cell line RAJI, which belongs to the acute B-lymphocyte leukemia group. It was found that YLS004 was significantly less toxic to RAJI than to the acute T-lymphocyte leukemia cell line.
2.Thank you for your inquiry. Currently, YLS010 is in the early stages of study. The clinical mode of administration will be intravenous, and the first category in the long-term toxicity guidelines, clinical single-agent intravenous administration for 14 days, has been selected. The results of this long-term toxicity trial can inform the design of clinical dosages and provide support for clinical trials and manufacturing.
3.Thanks for the question. As you said, we observed high toxicity of YLS010 to liver, kidney and lungs in our long-term toxicity experiments in rats, but we put the rats to death immediately after 14 consecutive days of administration, so there is no experimental evidence to show whether this toxicity is reversible or not, but your question he has also given us ideas to open up the follow-up experiments, and the results of the current toxicity experiments are frustrating, and we will explore the reversibility of the toxicity in follow-up experiments, to provide experimental evidence to show whether YLS010 can be entered into the next stage at a larger range of safe dosages.
4.Thanks for the question. Throughout the study, we attempted to obtain clinical samples of acute T-lymphoblastic leukemia. Fortunately, we were able to isolate the acute T-lymphoblastic leukemia tumor cells from a bone marrow aspirate obtained from a patient at the clinic. The article reports that YLS010 administration resulted in significant activity, surpassing that of YLS004 and the clinical drug Nelarabine, as shown in Figure 2B. Clinical samples from patients with acute T-lymphoblastic leukemia are still being collected, which may take some time.
Reviewer 3 Report
Comments and Suggestions for Authors
The authors have developed a new compound, YLS010, that is active against both Trx-1 and TERT in T-ALL cell line models. They have provided some evidence for YLS010 having similar mechanisms of action as their previously-reported analog, YLS004, and determined the cell death effects (apoptosis and ferroptosis), tolerated doses in rats, and efficacy in a xenotransplant model in mice. The manuscript could be improved in some regards:
Major criticisms
1) In the Introduction, and again in section 3.3 of the Results, it is stated that YLS004 has dual inhibitory activity on Trx-1 and TERT. The stated mechanism for these effects is biochemical inhibition of the pathways involved, i.e., oxidization of the sulfhydryl group in Trx-1, rendering it inactive, and incorporation of the metabolite 6-thio-in the telomerase synthesis process to inhibit its activity. It is also stated that due to structural similarity, it is suspected that YLS010 will have these same effects. However, to confirm that suspicion, the authors measured the effect of YLS010 on transcription and protein expression levels of Trx-1 and TERT in MOLT-4 and HUT-78 cells. That would only be logical if it was suspected that YLS010 acts by reducing transcription, translation, and/or stability of those proteins; instead, given the mechanisms proposed earlier, what is needed are assays for the redox state or activity of Trx-1, and for telomerase activity (such as telomere length or TRAP assay). Panel A of Figure 3 shows that YLS010 reduces levels of mRNA for Trx-1 and TERT; this is unlikely a non-specific effect of the toxicity of YLS010 to cells, since exposure was for only 12 hours, but it could be a secondary effect. It is not stated how long was the exposure at which data for panels C and D were obtained. Only panel B is relevant, because it is an assay for Trx-1 activity, but it may reflect only non-specific or secondary effects, since the data were obtained after exposure for 12 hours.
Experiments for Figure 3 should be replaced in the following way:
A. Only assays for activity, including one(s) relevant to telomerase, should be used
B. Assays should be performed within a few hours of drug exposure, because the suspected mechanism(s) of YLS010 are ones that should take effect quickly.
C. At the time point used, there should be documentation that there are only mild reductions in the number and viability of cells.
2) In panel E of Figure 8, the YLS010 dose level of 8 mg/Kg produced a markedly lower ratio of spleen weight to body weight, as compared to the other dose levels. This is interpreted to be evidence of better control of the transplanted T-ALL cells, which may be true. However, as an additional or alternative explanation, it may indicate that the drug has caused atrophy of the normal spleen, perhaps including the lymphoid element. To address these possibilities, the authors should compare the absolute spleen weight in 8 mg/Kg-treated mice to that of normal mice.
3) The rationale for why the authors created YLS010 is not clear. It is also not clear why the authors chose YLS010 over YSL004 for their studies, since the statement on lines 304-5 that “YLS010 had a much stronger anti-tumor effect on T-ALL cells compared to YLS004” seems to be an exaggeration. Is it possible that YLS004 might have a larger therapeutic window? The authors should comment more directly on their preference for YLS010.
Minor criticisms
1) The English needs some minor editing throughout, for errors including these:
A. Early in the Introduction, there are the following sentences:
“The clinical treatment of T-ALL currently relies on traditional chemotherapy modalities [1-3]. However, these modalities have poor sensitivity to current chemotherapeutic drugs and easily develop drug resistance [4,5].”
The second sentence is logically incorrect, because poor sensitivity and resistance are features of T-ALL, not of “modalities”. The second sentence should be corrected.
B. Line 46: “highly” should be “high”
C. Line 58: “maintaining of” should be “maintaining” or “maintenance of”
D. Line 65: “and PX-12” should be “PX-12”
E. The sentence beginning on Line 72 occurs 3 times; it should only occur once.
2) The units of the X-axis of Figures 1B and 1C should be specified. Presumably the values refer to micromolar concentration. In addition, it should be stated that the Y-axis values are the ratio of viable cells (as determined with CCK-8) as compared to no-drug values.
3) Lines 329-332 must be removed. Those are instructions to authors, and should not be included in an attempt to justify the way in which they have reported the results.
4) Line 350: “decrease in cell membrane potential” should be “decrease in mitochondrial membrane potential”
Comments on the Quality of English LanguageSee above
Author Response
1.Thank you for your suggestion. As you mentioned, YLS010 inhibits Trx-1 and TERT activity through biochemical pathways. It is important to determine if YLS010 has a sustained inhibitory effect on Trx1 activity, despite the rapid disulfide bond reduction process. Therefore, we measured the inhibition of TrxR/Trx1 activity after 48 hours of YLS010 treatment. To ensure accurate results and exclude false positives caused by decreased cell viability, we compared the ratio between TrxR/Trx1 and protein content. This allowed us to derive rigorous experimental results. To investigate the effect of YLS010 on Trx1 activity, we measured the mRNA content and protein expression of Trx1 after YLS010 treatment.
Wisman analyzed the mRNA expression of hTERT in human ovarian cancer and confirmed the presence of both hTERT and telomerase activity in malignant tumors with a similar expression pattern. Telomerase activity changes were characterized by measuring TERT transcription and translation using RT-qPCR and Western blotting, supported by relevant literature.
- G Bea A Wisman; Harry Hollema; Marco N Helder; Ageeth J Knol; Gretha T Van der Meer; Mindert Krans; Steven De Jong; Elisabeth G E De Vries; Ate G J Van der Zee. Telomerase in relation to expression of P53,c-myc and estrogen receptor in ovarian tumors. J. Oncol.2003,23,1451-1459.
- Zhang; J. Zhou; Q. Ye; K. Zeng; J. Pan; L. Chen; Y. Wang; B. Yang; Q. He; J. Gao; et al. 6-Dithio-2'-deoxyguanosine analogs induce reactive oxygen species-mediated tumor cell apoptosis via bi-targeting thioredoxin 1 and telomerase. Toxicol. Appl. Phaemacol.2020, 401, 115079.
- Chen W W; Xiong X X; zhou h y. Expression of telomerase activity,telomerase RNA component and telomerase catalytic subunit gene in lung cancer. Med. J.2002,115,290-292.
- Sheng W Y,Chien Y L,The dual role of protein kinase C in the regulation of telomerase activity in human lymphocytes. FEBS Letters2003,540,91-95.
- Sagawa Y; Nishi H; Isaka K. The correlation of TERT expression with c-myc expression in cervical cancer. Cancer Letters2001,168,45-50.
2.Thank you for your question, for the possible toxic injury of YLS010, we have explored it on both mouse and rat species, and we did not observe any significant spleen toxicity in both acute and long term toxicity experiments, so we believe that the dose of 8mg/kg is in the safe range of administration, and the recordings of body weights of the mice demonstrated the same result.
3.Thank you for your question, we have compared the toxicity between YLS010 and YLS004 several times in human acute T-lymphoblastic leukemia cell lines, and the activity is stronger than YLS004 in both cases, and what's more, we have been trying to collect clinical samples from patients with acute T-lymphoblastic leukemia throughout the study, but unfortunately we only obtained a single clinical sample, but the activity of YLS010 against this wild-type human Acute T-lymphoblastic leukemia was much more active than YLS004 and Nelarabine, which made us think that YLS010 seems to have more potential than YLS004 for clinical application, so we chose YLS010 for the follow-up study, and it is true that YLS010 showed a stronger toxicity in the follow-up, but this is not a reason for us to completely abandon YLS010 as a candidate compound. We will subsequently try to construct a PDX model to further investigate the effect of YLS010. Unfortunately, we have not obtained enough clinical samples so far.

Round 2
Reviewer 1 Report
Comments and Suggestions for Authors
The authors revised the manuscript according to my observations.
Concerning the WB analysis in fig.4G and also K, as shown in the supplementary data file, the membrane relative to Caspase3 detection was cut at 25kD level and the two parts incubated separately with the same anti Caspase 3 antibody. I don't understand why the authors made this decision. The time of exposure by the “chemiluminescence instrument” (not better defined in MM) could be lengthened If the bands of cleaved caspase 3 were faint.
However, since the membrane has been cut, the upper panels of fig4G and K should be presented as two independent panels.
Author Response
Image has been replaced. Thank you to reviewer 1 for bringing this to our attention!
Because the gray value of pro-caspase3 was found to be too high in the pre-experiment and cleaved-caspase3 could not be observed when exposed on a membrane strip, the formal experiment was performed by dividing this caspase3 protein gel into two parts and incubating them separately. In addition, the exposure of the protein strip of cleaved-caspase3 was the result of an extended exposure time.

Reviewer 3 Report
Comments and Suggestions for Authors
The authors have made some changes to my previous criticisms, but the response is not sufficient. Responses are deficient to:
Major criticism #1. Given that the proposed mechanism of YLS010 is inhibition of the activities of Trx-1 and TERT, it is not relevant to show that the levels of those proteins is decreased, and at a relatively late point (48 hours). The authors' citations of an equivalence between TERT protein level and TERT activity are not convincing. If the authors still refuse to assay the activities at an early time point as recommended, the findings of Figure 3 can only be cited as possible additional contributors to the effects of YLS010, and any claims as to effects on the activities of Trx-1 and TERT can only be based on an assumption of similarity to YLS004.
Major criticism #2. The authors have not addressed the question. It is simply sufficient to declare whether the spleen weight at 8 mg/Kg is within the range of normal, or is below normal.
Minor criticism #1E. The sentence in question still occurs 3 times within a short span of the manuscript.
Comments on the Quality of English LanguageNothing to add. Editorial staff should review.
Author Response
1.Answer:Thank you for your inquiry. After conducting further literature research and internal discussions, we have decided to remove the section discussing the dual inhibition of Trx1 and TERT in YLS010 from the article. We will re-experimentally justify this section in a follow-up study. Your question has helped us to conduct the follow-up study more rigorously. Thank you again for your contribution.
In addition, we still believe that the detection of TrxR/Trx1 and TERT activities in a short period of time has no obvious significance, and the key to the tumor cell killing effect of YLS010 lies in its ability to inhibit TERT and TrxR/Trx1 activities in tumor cells for a long period of time, and the inhibitory effect of a short period of time does not significantly contribute to its tumor cell killing effect.
2.
Answer:Thank you for reminding us that the normal physiological weight of the spleen of this species of mice obtained from the animal company is usually around 100mg, and according to our raw data, the spleen weight of the mice in the 8mg/kg dose group was 0.12±0.01g, so there was no shrinkage of the spleen in the 8mg/kg dose group.
3.
Answer:Thank you for your reminder, we rechecked the article, and the sentence in line 72 that you said was repeated three times was "PX-12 has demonstrated a promising safety profile and good tolerability in phase I clinical trials in advanced solid tumors [14]. PX-12 has demonstrated a promising safety profile and good tolerability in phase I clinical trials in advanced solid tumors [14]. Although its phase II clinical trial ended in failure due to the lack of significant clinical efficacy, its potential anti-tumor efficacy is worth exploring in depth [15]." This sentence? We did not find the sentence repeated three times, and would appreciate it if you could point out the duplicates if possible, thank you!

Round 3
Reviewer 3 Report
Comments and Suggestions for Authors
The authors have made satisfactory changes to their manuscript. In regards to points in the most recent review:
Point #2: Now that the authors have confirmed that spleen weight in the 8mg/kg dose group is close to normal, certainly not lower than normal, I recommend that the manuscript be slightly altered to include this information; otherwise, a reader might have the same question that I did. This would be easily done; e.g., simply modify the end of the sentence on line 476, from the present "indicating a better treatment effect" to "indicating a better treatment effect, and spleen weight was essentially normal."
Point #3: The text in question begins on line 73 in the latest version of the manuscript. I have only modified it here by inserting numbers at the beginning of each repetition of the underlined sentence:
"(1) The process involves reverse transcription of a single strand of DNA, followed by hydrolysis of the RNA component in the heterozygous double strand using RNA hydrolase. (2) The process involves reverse transcription of a single strand of DNA, followed by hydrolysis of the RNA component in the heterozygous double strand using RNA hydrolase. This leaves behind the primer fragments, which are then used in Semi-conservative replication to form DNA double strands, ultimately achieving telomere lengthening [17,18]. (3) The process involves reverse transcription of a single strand of DNA, followed by hydrolysis of the RNA component in the heterozygous double strand using RNA hydrolase. In normal human cells, telomeres gradually shorten with each cell division."
A properly-revised version of this text would keep only the first occurrence of this sentence:
"The process involves reverse transcription of a single strand of DNA, followed by hydrolysis of the RNA component in the heterozygous double strand using RNA hydrolase. This leaves behind the primer fragments, which are then used in semi-conservative replication to form DNA double strands, ultimately achieving telomere lengthening [17,18]. In normal human cells, telomeres gradually shorten with each cell division."
Author Response
Answer:Thank you very much for your careful and serious guidance on both the research mechanism and the writing of the thesis, I have benefited a lot from this process, and I believe that these suggestions of yours will play a very important role in my future research, thanks again! The following is a response to your two suggestions!
